# Effect of sanitation improvements on soil-transmitted helminth eggs in courtyard soil from rural Bangladesh: Evidence from a cluster-randomized controlled trial

**Laura H. Kwong** [1¤a]*, **Debashis Sen** [2], **Sharmin Islam**[2], **Sunny Shahriar**[2], **Jade Benjamin-Chung** [3¤b], **Benjamin F. Arnold** [4], **Alan Hubbard**[3], **Sarker Masud Parvez**[2], **Mahfuza Islam** [2], **Leanne Unicomb** [2], **Md. Mahbubur Rahman**[2], **Kara Nelson**[5], **John M. Colford, Jr.** [3], **Stephen P. Luby**[1], **Ayse Ercumen**[6]

1 Woods Institute for the Environment, Stanford University, Stanford, California, United States of America, 2 International Centre for Diarrhoeal Disease Research, Bangladesh (icddr,b), Dhaka, Bangladesh, 3 School of Public Health, University of California, Berkeley, California, United States of America, 4 Francis I. Proctor Foundation, University of California, San Francisco, San Francisco, California, United States of America, 5 Department of Civil Engineering, University of California, Berkeley, California, United States of America, 6 Department of Forestry and Environmental Resources, North Carolina State University, Raleigh, North Carolina, United States of America

¤a Current address: School of Public Health, University of California, Berkeley, California, United States of America
¤b Current address: Division of Epidemiology and Population Health, Stanford University, Stanford, California, United States of America
* kwong.laura@gmail.com

## Abstract

Improved sanitation has been hypothesized to reduce soil-transmitted helminth (STH) infections by reducing the prevalence and concentration of STH eggs/larvae in soil. We evaluated the effect of a randomized sanitation program (providing households with an improved dual-pit latrine, tools for child/animal feces management, and behavioral messaging) on reducing the prevalence and concentration of STH eggs in soil from household courtyards. We collected soil samples from 1405 households enrolled in the sanitation intervention (n = 419) and control (n = 914) groups of a cluster-randomized controlled trial (WASH Benefits) in rural Bangladesh approximately 2 years after the initiation of the interventions. We analyzed samples for *Ascaris lumbricoides*, *Trichuris trichiura*, and hookworm eggs by microscopy. We estimated prevalence ratios (PR) and egg count ratio (ECR) to compare the prevalence of STH eggs and arithmetic and geometric mean egg counts for STH eggs per gram of soil in the sanitation and control arms. Among intervention households, latrines achieved high and sustained user uptake by adults while child open defecation remained common and most households did not dispose of child feces hygienically. In courtyard soil from control households, the prevalence of any STH eggs was 75.7% and the prevalence of any larvated STH eggs was 67.3%. *A. lumbricoides* was detected in 63.0% of control samples and *T. trichiura* in 55.7% of control samples; hookworm was not detected in any sample. In the control arm, the arithmetic mean egg count for any STH was 3.96 eggs/dry gram, while the geometric mean was 1.58 eggs/dry gram. There was no difference between the

**Data Availability Statement:** The dataset used for analysis is available on the OSF website: https://osf.io/6u7cn/.

**Funding:** This study is based upon work supported by the Bill &Melinda Gates Foundation (OPPGD759) (https://www.gatesfoundation.org/) and the National Institutes of Health (R01HD078912) (www.nih.gov) through grants to JMC. LHK was a recipient of the Stanford Wood's Institute for the Environment Goldman Graduate Fellowship and the National Science Foundation Graduate Research Fellowship under Grant No. DGE-114747 (https://www.nsfgrfp.org/). Any opinion, findings, and conclusions or recommendations expressed in this material are those of the authors(s) and do not necessarily reflect the views of the National Science Foundation. The funders had no role in the study design, data collection and analysis, decision to publish, or preparation of the manuscript.

**Competing interests:** The authors have declared that no competing interests exist.

intervention and control groups in the prevalence of any STH eggs (PR = 0.98 (95% CI: 0.91, 1.05)) or mean egg counts (ECR = 0.08 (95% CI: -0.10, 0.26) for geometric mean and 0.07 (95% CI: -0.22, 0.37) for arithmetic mean). Adjusted models gave similar results. A compound-level sanitation intervention that provided improved latrines and tools for disposal of child and animal feces did not have an impact on STH eggs in soil. In order to effectively reduce the prevalence and concentration of STH eggs in the environment, sustained, widespread use of sanitation strategies to isolate and hygienically dispose of child and animal feces may need to complement traditional strategies for containment of adult human feces.

  **Trial Registration:** NCT01590095.

## Author summary

Improved sanitation has been hypothesized to reduce soil-transmitted helminth (STH) infections by reducing the prevalence and concentration of STH eggs/larvae in soil. We evaluated the effect of a randomized sanitation program (providing households with an improved dual-pit latrine, tools for child/animal feces management, and behavioral messaging) on reducing the prevalence and concentration of STH eggs in soil from household courtyards. We collected soil samples from 1405 households enrolled in the control and sanitation intervention arms of a cluster-randomized controlled trial (WASH Benefits) in rural Bangladesh approximately 2 years after the initiation of the interventions. We analyzed samples for *Ascaris lumbricoides*, *Trichuris trichiura* and hookworm eggs by microscopy. We found no effect of the sanitation intervention on STH eggs in soil. In order to effectively reduce the prevalence and concentration of STH eggs in the environment, sustained, widespread use of sanitation strategies to isolate and hygienically dispose of child and animal feces may need to complement traditional strategies for containment of adult human feces.

## Introduction

Soil-transmitted helminth (STH) infections affect more than 1.5 billion people worldwide [1]. These infections include *A. lumbricoides* (roundworm), *T. trichiura* (whipworm), and *Necator americanus*, *Ancylostoma ceylanicum*, and *Ancylostoma duodenale* (hookworm) infections. The principal approach to mitigating the impact of STH infections is mass drug administration, often delivered through school-based programs or integrated into vaccination programs [2]. Anthelmintic benzimidazoles drugs such as albendazole and mebendazole are effective in reducing STH infection [3,4]. The WHO recommends annual mass drug administration in areas where STH prevalence is 20% or more because rapid reinfection following treatment is common [5]. A systematic review and meta-analysis has shown that, 12 months post-treatment, the infection prevalence can revert to 94% of pre-treatment levels for *A. lumbricoides*, 82% for *T. trichiura* and 57% for hookworms [6]. The frequent and widespread use of anthelmintic drugs may result in the emergence of drug resistance, which would substantially reduce the effectiveness of the limited number of drugs currently available for treatment of STH infections [2].

  Sanitation systems that isolate human feces from the environment should prevent potential new hosts from ingesting STH eggs from the feces of infected individuals. Hence, sanitation improvements that interrupt environmental transmission cycles of STH may be critical to sustainably reduce the global burden of STH infections [7,8]. Two systematic reviews have found that households that reported access to and use of any sanitation facility had approximately

half the odds of STH infection compared to households with no reported availability or use of sanitation facilities [9,10]; however, this evidence was mostly based on observational studies of low quality. In rural India, Bangladesh, Kenya and Timor-Leste, randomized controlled trials of sanitation interventions providing on-site latrine access, alone or combined with water and hygiene interventions, have either found no effect on STH [11–13] or reductions in some but not other STH species [14–16]. In contrast, a study in urban Brazil found that increasing household connections to sewerage halved the prevalence of STH infections [17]. Sanitation interventions are hypothesized to reduce STH infection through reducing the prevalence and/ or concentration of STH eggs in the environment. However, there are little empirical data on the impact of sanitation interventions on this intermediate outcome [18].

The impact of sanitation interventions on environmental transmission of STH may be assessed by quantifying STH eggs in the soil, a necessary component of the STH life cycle. Unembryonated eggs of all three species are passed in stool and incubate in soil. *A. lumbricoides* and *T. trichiura* eggs can survive in soil for months, becoming infective in 1–15 weeks [19]. Hookworm eggs, on the other hand, quickly hatch in soil; the larvae that hatch seek moisture and can survive for weeks and become infective within 2 weeks [19]. As such, it is more likely to find *A. lumbricoides* and *T. trichiura* eggs than hookworm eggs in the soil. Egg survival is longer in warm and moist environments without direct sunlight [19,20].

Methods exist to enumerate STH eggs in soil, biosolids, and water, and on produce and vegetation [20]. STH eggs have been detected in outdoor soil in various settings in South/Southeast Asia [21–28], South America [29] and sub-Saharan Africa [30,31]. STH eggs have also been detected on floors inside homes in the Philippines [27] and in classrooms with soil floors in Nigeria [31]. These studies were mostly descriptive, and sample sizes were relatively small, ranging from 15 [29] to 330 [28] people. One study assessed the effect of a randomized sanitation intervention on STH eggs in environmental reservoirs in rural Kenya and found no effect [18].

Here, we examine the effect of a randomized sanitation intervention on the presence and concentration of STH eggs in courtyard soil in rural Bangladeshi households. We conducted this study among a subset of households enrolled in a cluster-randomized controlled trial (WASH Benefits Bangladesh, ClinicalTrials.gov identifier NCT01590095) that measured the impact of individual and combined water, sanitation, hygiene and nutrition interventions on linear growth, cognitive development, and enteric disease, including STH infections, in young children [32]. The trial also measured pathogens and indicators of fecal contamination in the environment in a subset of enrolled households as intermediate outcomes between the interventions and child health outcomes.

## Methods

### Ethics statement

Participants provided written informed consent in the local language. The study protocol was approved by the Ethical Review Committee at The International Centre for Diarrhoeal Disease Research, Bangladesh (PR-11063), the Committee for the Protection of Human Subjects at the University of California, Berkeley (2011-09-3652), and the institutional review board at Stanford University (25863).

### Study setting and design

In Bangladesh, the WASH Benefits trial enrolled participants in four rural districts. Eight pregnant women in their second or third trimester who lived in nearby households were grouped into clusters and eight adjacent clusters were grouped into a block. Clusters were block-

randomized into one of six intervention arms or a double-sized control arm. The intervention arms consisted of single and combined water, sanitation, hygiene, and nutrition interventions. This study involved households in the control and sanitation arms of the WASH Benefits trial.

The prevalence of STH infection among children aged 2–12 years old in the control arm was 36.8% for *A. lumbricoides*, 9.2% for hookworm, and 7.5% for *T. trichiura* [16]. Bangladesh has conducted school-based deworming for school-aged children and immunization-based deworming for pre-school-aged children with mebendazole two times per year since 2008 [33]. A 2010 evaluation of the 2009 mass drug administration campaign in two districts of Bangladesh found that for children aged 5–14 years old treatment coverage for one of the twice-annual campaigns was 46–54% depending on district and campaign month [33]. Among children aged 4–12 years old enrolled in WASH Benefits, 74% of children in the sanitation arm and 70% of children in the control arm were reported to have received deworming within the last six months [16].

### Intervention components

The compound-level sanitation intervention included upgrading or replacing existing latrines with improved latrines that were lined with concrete rings and had two pits so that after the first pit filled, the superstructure could be moved to a second pit while the contents of the first pit were left to undergo natural inactivation of pathogens such as STH eggs. The intervention also included potties for young children, and sani-scoop hoes to remove animal and child feces from households and courtyards. Households in Bangladesh are clustered in multi-family compounds that share a central courtyard; each household in compounds with study households received a latrine upgrade, potty and sani-scoop. Locally recruited community health promoters visited households in the intervention group at least twice a month to deliver sanitation-related behavior change communication focused on safe management of human and animal feces. Promoters did not visit control households.

### Intervention uptake

Structured observations of intervention uptake were conducted by WASH Benefits staff members as part of the intervention uptake assessment, approximately 15 months after the intervention delivery and 9 months before our STH assessment [34]. At this timepoint, adults were observed to use a hygienic latrine, defined as a latrine with a functional water seal and no feces visible on the latrine slab or floor, in 94% of sanitation-arm households and 40% of control-arm households [34]. However, open defecation by children and unhygienic child feces management remained prevalent. The percentage of children reported to defecate in the open was 81% among children 0–2 years old, 37% among children 3–7 years old, and 9% among children 8–14 years old in the sanitation arm; prevalence of open defecation was similar among children in the control arm [16]. Open defecation by children commonly occurred in the compound courtyard and child feces were typically disposed of unhygienically, *i.e.*, thrown into bushes, open waste heaps and drains, or left on the ground [35]. Households were observed to unhygienically dispose of child feces in 64% of sanitation-arm households and 84% of control-arm households [34]. Hence, human feces were observed in 21% of sanitation-arm compounds and 30% of control-arm compounds [34]. Further details of the trial design, intervention delivery, and uptake assessments have been described elsewhere [34,36–38].

### Soil sample collection and analysis

Soil was collected from a random subset of households enrolled in the sanitation or control arms of the WASH Benefits Bangladesh trial from May 2015 to April 2016. From 1991–2020,

the average rainfall ranged from a low of 7.4 mm in January to 498.0 mm in August [39]. Data collection followed the order of intervention roll-out among participating households so that samples were collected approximately 24 months after initiation of interventions in each household. Data collection spanned one year, allowing for comparison of samples collected in the wet and dry seasons.

Field staff sampled courtyard soil from the area just in front of the entrance of the enrolled household [18]. Previous work has found STH eggs concentration in soil to be similar in front of the household entrance and the latrine entrance [30]. Field staff marked a 30 x 30 cm area using an ethanol-disinfected metal stencil and used an ethanol-disinfected metal trowel to scrape the area once horizontally and once vertically to collect approximately 50 g of topsoil in a sterile Whirlpak bag. Samples were transported on ice to the field laboratory of the International Centre for Diarrhoeal Disease Research, Bangladesh and processed within one day to quantify eggs of *A. lumbricoides*, *T. trichiura*, and hookworm using a protocol adapted from the USEPA method for enumerating *A. lumbricoides* eggs in fecal sludge [40,41]. The adapted protocol demonstrated a recovery efficiency of 73% for *A. lumbricoides* eggs in laboratory experiments [41]. As the method was optimized to detect *A. lumbricoides*, the recovery efficiencies for other STH species were likely lower.

In brief, a 15 g soil aliquot was soaked overnight in 1% 7X detergent solution (MP Biomedicals, Irvine, CA), then hand-shaken for 2 minutes and vortexed on 2000 rpm for 15 seconds to dislodge STH eggs from soil particles. The solution was poured through a 50-mesh sieve to remove large soil particles. The supernatant was left to settle for 30 minutes and then aspirated without disturbing the soil at the bottom. Approximately 40 mL of 1% 7X solution was added to the precipitate and the solution was centrifuged at 1000 g for 10 minutes; the supernatant was discarded. 5 mL of zinc sulfate flotation solution (1.25 specific gravity) was added to the precipitate, vortexed for 30 seconds, centrifuged at 1000 g for 5 minutes and the supernatant was saved; this procedure was conducted a total of three times. The combined supernatant from the three flotation steps was filtered through a 500-mesh sieve to capture STH eggs. The sieve was rinsed into a Falcon tube using distilled water, the rinse water was centrifuged at 1000 g for 5 min, and the supernatant was removed with a pipette until there was 1 mL left at the bottom of the tube. 25 mL of 0.1 N sulfuric acid solution was added to the tube. The tube was capped loosely and incubated at 28˚ C for 28 days to allow viable eggs to develop larvae. At the end of the incubation period, the solution was centrifuged at 1000 g for 3 min and aspirated to a final volume of 1 mL. The 1 mL solution was transferred to a Sedgewick-Rafter slide and examined under the microscope for *A. lumbricoides*, *T. trichiura*, and hookworm eggs using a visual identification chart to distinguish the type of egg and whether it was larvated or non-larvated [41]. The numbers of larvated and non-larvated eggs for each species were recorded separately to differentiate viable and non-viable eggs. An additional 5 g soil aliquot was oven-dried overnight to determine moisture content and dry weight.

For quality control, a laboratory blank was processed once every other day by repeating the protocol without a soil sample. 10% of samples were processed twice, with the second sample referred to as the technical replicate. 10% of samples were counted by two independent analysts to assess interrater reliability. Additionally, for each sample, lab technicians took a picture of the first occurrence of each type of egg (larvated *A. lumbricoides*, non-larvated *A. lumbricoides* etc.); the pictures were reviewed for accuracy of categorization by study investigators (LHK, AE).

### Minimum detectable effect size

Internal pilot data from the first 100 samples in the control arm found that the prevalence of eggs in soil was 67% for *A. lumbricoides*, 36% for *T. trichiura* and 78% for either one of the

species; hookworm was not detected, potentially because fragile hookworm eggs degraded while the sample was soaked in detergent overnight or during other processing steps. The pilot data were used to calculate intra-class correlation coefficients (ICC) by geographical area of 0.05 for *A. lumbricoides* and 0.26 for *T. trichiura*. Based on the daily processing capacity of the lab, we aimed to sample 1500 compounds. This sample size allowed a minimum detectable relative reduction in prevalence between arms (with 80% power and a two-sided alpha of 0.05) of 13% for *A. lumbricoides*, 33% for *T. trichiura* and 10% for any STH.

### Statistical analysis

Full replication of blinded analysis was conducted by two authors (LHK and AE) according to the pre-specified analysis plan (https://osf.io/6u7cn/). We calculated the prevalence of total and larvated eggs in soil for each species (*A. lumbricoides*, *T. trichiura*, hookworm), for any STH, and for multiple species of STH. We calculated the concentration of STH in soil for each species and for any STH. Prevalence was defined as the presence of STH eggs of the specified species, and concentration was defined the number of eggs of the specified species per dry gram of soil.

We estimated prevalence differences (PD) and prevalence ratios (PR), as well as relative egg count ratio (ECR) between the sanitation and control arms. PD was defined as the difference in the prevalence and PR as the ratio of the prevalence in the sanitation vs. control arm. ECR was defined as the ratio of egg counts in the sanitation vs. control arm minus 1 such that values <0 indicate a reduction in the sanitation arm. We calculated ECRs using both arithmetic and geometric means. Arithmetic means are more sensitive to high infection intensities that are associated with higher morbidity burden and higher likelihood of transmission; geometric means are less sensitive to extreme data points that skew arithmetic means [4,42,43]. Geometric means were calculated by taking the natural log of the egg count values after replacing counts of 0 eggs per gram [epg] with 0.5 epg. Technical replicates (the second aliquot of a sample analyzed) were dropped from the analysis. We estimated unadjusted and adjusted intention-to-treat effects using the R (version 3.3.2) package for targeted maximum likelihood estimation with SuperLearner; SuperLearner is an algorithm that uses cross-validation to test and weight multiple machine learning models to optimize predictive accuracy [44,45]. We used robust standard errors that accounted for clustering. We defined a result as significant if the corresponding 95% confidence intervals did not cross 1 for prevalence and egg count ratios or 0 for prevalence differences.

As randomization led to very good baseline balance of sociodemographic and water, sanitation, and hygiene characteristics across arms [36] (Table 1), bias due to confounding due to these factors is unlikely. Additionally, adjustments are unlikely to substantially improve the precision of binary outcomes such as prevalence [46]. Hence, we report unadjusted effect estimates as our primary parameters. Secondary adjusted analyses accounted for the lab staff member who processed or counted the sample, sampling month, soil moisture content, mother's education level (stratified into no education, primary or secondary), degree of household food insecurity, number of children <18 years in the household, total number of individuals living in the compound, reported time to the household's primary drinking water source, housing materials (floor, walls, roof), presence of electricity, and household assets (a table, chair, wardrobe, bed, stool, clock, radio, landline phone, mobile phone, working black/white or color TV, working refrigerator, adult bicycle, motorcycle, sewing machine). Covariates were included in the adjusted analysis if bi-variate models using the likelihood ratio test indicated that the covariate was associated with the outcomes (p<0.2). Categorical covariates that occurred in <5% of the sample were excluded from models.

**Table 1. Household-level enrolment characteristics for households that participated in assessment of soil-transmitted helminths in soil, by study arm.**

| Characteristic | Control (N = 914) | Sanitation (N = 491) |
|---|---|---|
| Number of household members | 4.7 (2.2) | 4.7 (2.1) |
| Number of household members <18 years old | 1.6 (1.2) | 1.6 (1.2) |
| Number of compound members | 11.1 (6.2) | 11.2 (6.7) |
| Mother's age (yr) | 23.9 (5.0) | 24.0 (5.0) |
| Mother's education | | |
| No education | 133 (14.6%) | 74 (15.1%) |
| Primary (1-5y) | 294 (32.2%) | 161 (32.8%) |
| Secondary (>5y) | 487 (53.3%) | 256 (52.1%) |
| Mother's years of formal education | 5.9 (3.4) | 5.9 (3.4) |
| Father's years of formal education | 4.9 (4.0) | 5.0 (4.1) |
| Father works in agriculture | 288 (31.5%) | 158 (32.2%) |
| Improved roof material | 901 (98.6%) | 481 (98.0%) |
| Improved wall material | 672 (73.5%) | 364 (74.1%) |
| Improved floor material | 90 (9.8%) | 54 (11.0%) |
| Percent of children <3 years old reported practicing open defecation | 154 (80.6%) | 76 (81.7%) |
| Percent of children 3–8 years old reported practicing open defecation | 196 (40.2%) | 99 (37.9%) |
| Percent of men reported practicing open defecation | 65 (7.2%) | 35 (7.2%) |
| Percent of women reported practicing open defecation | 43 (4.7%) | 22 (4.5%) |
| Child feces observed in household courtyard | 17 (1.9%) | 3 (0.6%) |
| Adult feces observed in household courtyard | 88 (9.6%) | 38 (7.7%) |
| Household owns private latrine | 482 (52.7%) | 260 (53.0%) |
| Latrine has slab | 827 (94.7%) | 430 (91.5%) |
| Latrine has functional water seal | 229 (29.6%) | 118 (28.7%) |
| Latrine has observed feces on slab | 415 (48.1%) | 231 (50.9%) |
| Child potty is available | 30 (3.3%) | 19 (3.9%) |

We conducted the following pre-specified subgroup analyses, which account for climactic factors and the load of STH eggs potentially entering the environment: 1) wet season (Jun-Oct) vs. dry season (Nov-May); 2) number of individuals living in the compound (<10 vs. ≥10); 3) reported deworming of children in the compound within the prior 6 months (<2/3 vs. ≥2/3 dewormed); and 4) reported deworming of children in the cluster (8 nearby study compounds) within the prior 6 months (<2/3 vs. ≥2/3 dewormed). The cut-off values for number of individuals living in the compound and proportion of children dewormed correspond to the median of the empirical data across both arms.

## Results

### STH prevalence and concentration in soil

We sampled courtyard soil from 914 control and 491 intervention households. Randomization led to very good covariate balance across arms among the subset of sampled households (Table 1). Few households had animals that can spread STH by passage after consuming human feces (pigs, dogs) and/or are known hosts for the STH species we investigated (*A. lumbricoides* in pigs, *A. ceylanicum* in dogs and cats). During all 8 rounds of surveillance, at most 4/750 households (0.5%) owned one or more pigs and 8/750 households (1.1%) owned one or more cats or dogs. Within the control arm, the prevalence of total eggs (larvated or non-larvated) was 63.0% for *A. lumbricoides* and 55.7% for *T. trichiura*; hookworm was not detected

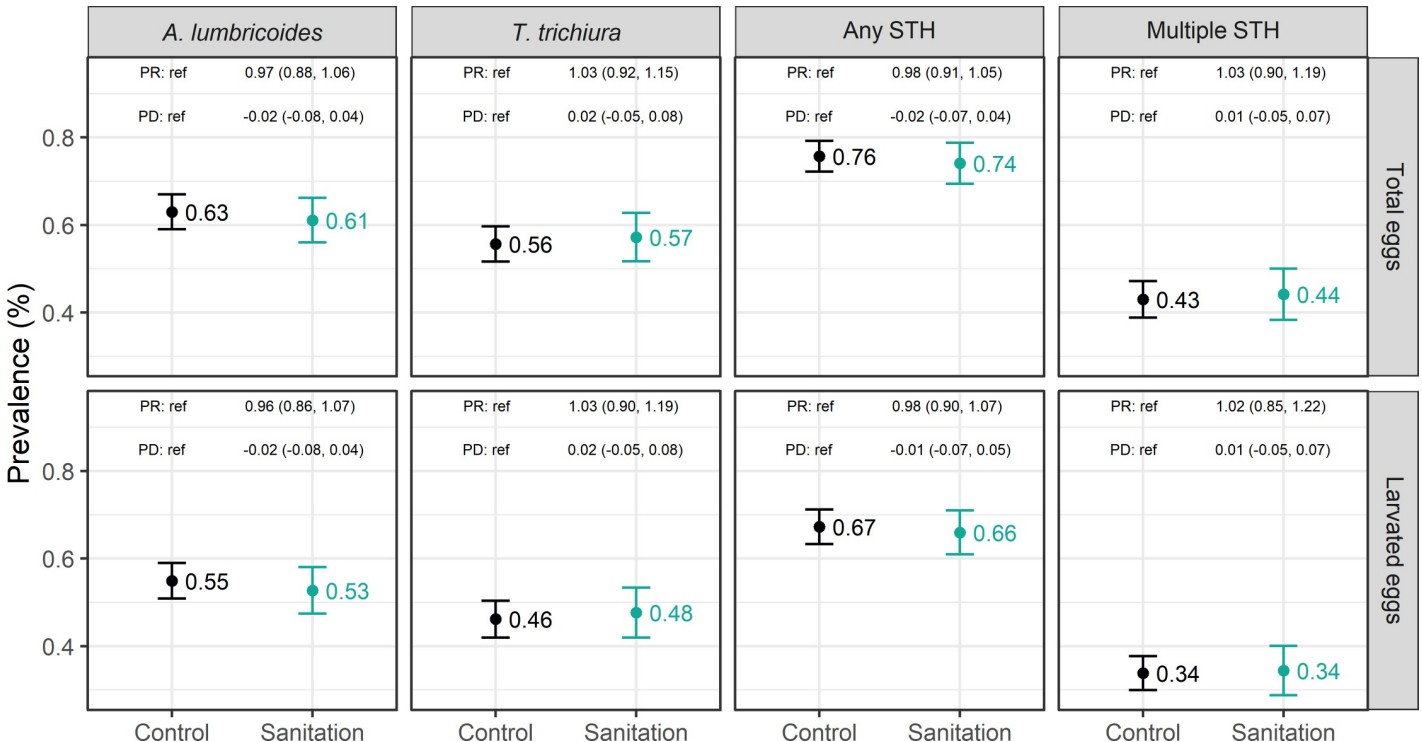

**Fig 1. Prevalence of total and larvated *A. lumbricoides* eggs, *T. trichiura* eggs, any soil-transmitted helminth eggs, and multiple species of soil-transmitted helminth eggs in soil in the control (n = 914) and sanitation (n = 491) arms.**

in any sample and therefore hookworm data is omitted from all presentations of results (Fig 1). Considering only *A. lumbricoides* and *T. trichiura*, the prevalence of any STH eggs was 75.7% and the prevalence of multiple species of STH eggs was 43.0%. The prevalence of larvated (*i.e.*, viable) STH eggs was 54.9% for *A. lumbricoides* and 46.2% for *T. trichiura* (Fig 1). The prevalence of any larvated STH eggs was 67.3% and the prevalence of multiple species of larvated STH eggs was 33.8%. The arithmetic and geometric mean counts of total and larvated eggs per gram dry soil are shown in Fig 2.

Quality control assessment found the two-way, mixed, agreement, single-measures intraclass correlation was 0.969 for samples processed in replicate and 0.995 for egg counts of the same slide enumerated by different technicians (excellent correlation). There was 0% contamination in blanks.

### Effect of the sanitation intervention on STH eggs in soil

The sanitation intervention had no significant impact on the prevalence in soil of total *A. lumbricoides* eggs (PR = 0.97 (95% CI: 0.88–1.06)), *T. trichiura* eggs (PR = 1.03 (95% CI: 0.92, 1.15)), any STH eggs (PR = 0.98 (95% CI: 0.91, 1.05) or multiple species of STH eggs (PR = 1.03 (95% CI: 0.90–1.19)) (Fig 1). Similarly, the sanitation intervention had no significant impact on the prevalence in soil of larvated *A. lumbricoides* eggs, larvated *T. trichiura* eggs, any larvated STH eggs, or multiple species of larvated STH eggs (Fig 1). The sanitation intervention also did not reduce the arithmetic or geometric mean egg count for *A. lumbricoides*, *T. trichiura*, or any STH for total or larvated eggs (Fig 2). Results from adjusted analyses were consistent with unadjusted analyses and showed no significant differences in prevalence or concentration of STH eggs between the intervention and control arms (Tables A and B in S1 Appendix).

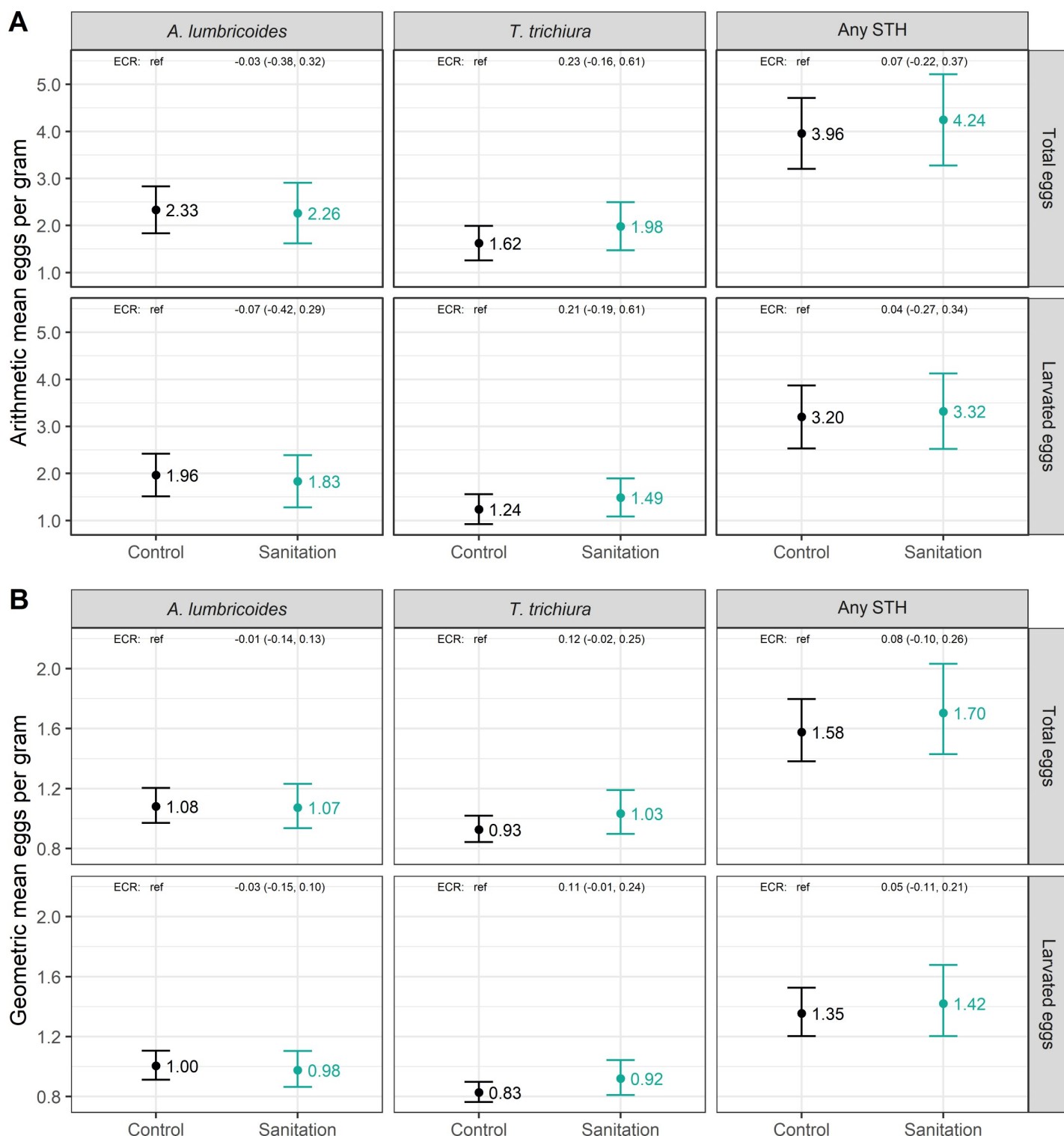

**Fig 2. Arithmetic and geometric counts of total and larvated** *A. lumbricoides* **eggs,** *T. trichiura* **eggs, and any soil-transmitted helminth eggs in soil in the control (n = 914) and sanitation (n = 491) arms.**

## Subgroup analyses

The prevalence of *T. trichiura* was significantly higher in the dry season (61% (95% CI: 57, 65)) compared to the wet season (40% (95% CI: 32, 49)) while the prevalence of *A. lumbricoides* was not significantly different between the seasons (Table C in S1 Appendix). There were no significant differences in the prevalence of larvated eggs across seasons. The concentration of STH eggs was also higher in the dry season than the wet season; the arithmetic mean egg count per gram dry soil for *A. lumbricoides* was 2.71 epg (95% CI: 2.06, 3.36) in the dry season and 1.20 epg (95% CI: 0.87, 1.53) in the wet season; for *T. trichiura* the mean egg count was 1.98 epg (95% CI: 1.51, 2.44) in the dry season and 0.57 epg (95% CI: 0.37, 0.77) in the wet season (Table D in S1 Appendix). Seasonal trends were similar for geometric means (Table D in S1 Appendix). Neither the prevalence nor the concentration of total or larvated eggs significantly differed by number of individuals in the compound, or reported deworming status of children in the compound or cluster (Tables C and D in S1 Appendix).

Subgroup analyses on the effect of the sanitation intervention by season, number of individuals living in the compound, and reported deworming status of children in the compound and cluster in the prior six months showed no significant differences between the intervention and control arms in prevalence or concentration of STH eggs in soil in any subgroup, consistent with the primary analysis using data from all samples (Figs 3, 4 and Fig A and Table E-H in S1 Appendix). We observed modest reductions of borderline significance in the prevalence of *A. lumbricoides* in the sanitation *vs.* control groups when ≥2/3 of children in the compound were dewormed (PR = 0.91 (95% CI: 0.79, 1.04)) or when ≥2/3 of children in the cluster were dewormed (PR = 0.89 (95% CI: 0.77, 1.02)).

## Discussion

We have demonstrated that nearly three-quarters (75.7%) of courtyard soil samples were contaminated with *A. lumbricoides* or *T. trichiura* eggs in rural Bangladeshi households, with two-thirds (67.3%) of samples contaminated with larvated eggs that are expected to be infectious. Among households in the sanitation arm, where adults regularly used hygienic latrines but only a third of households safely disposed of child feces [34,47], the prevalence and concentration of total or larvated *A. lumbricoides* or *T. trichiura* eggs in courtyard soil was not significantly lower compared to households in the control arm.

The prevalence of STH eggs we found in courtyard soil were on the higher end of estimates from other settings. In other studies, the prevalence of *A. lumbricoides* in soil ranged from 6% in Thailand [28] to 61% in Nigeria [31], and prevalence of *T. trichiura* in soil ranged from 6% in Nepal [25] to 41% in the Philippines [27]. While other studies have found that the prevalence of *A. lumbricoides* and *T. trichiura* in the soil was higher during the wet season than the dry season [25,27,28], we found that the prevalence of *A. lumbricoides* was not significantly different between seasons, and the prevalence and concentration of *T. trichiura* was higher during the dry season.

In our study's sister site (WASH Benefits Kenya), the prevalence of *A. lumbricoides* or *T. trichiura* in soil in front of households was 19% using the same sampling and enumeration protocol [18]. The geometric mean of *A. lumbricoides* or *T. trichiura* eggs in our study (2 epg) was also higher than what was observed in Kenya (0.05 epg). This is consistent with the higher prevalence of STH infection in the study population in Bangladesh vs. Kenya and with the higher population density of Bangladesh, which would be expected to provide more opportunities for environmental contamination with STH eggs. In Bangladesh, 37% of children enrolled in the WASH Benefits trial were infected with *A. lumbricoides* and 8% with *T. trichiura*, with the geometric mean fecal egg count of 5.2 epg for *A. lumbricoides* and 0.4 epg for

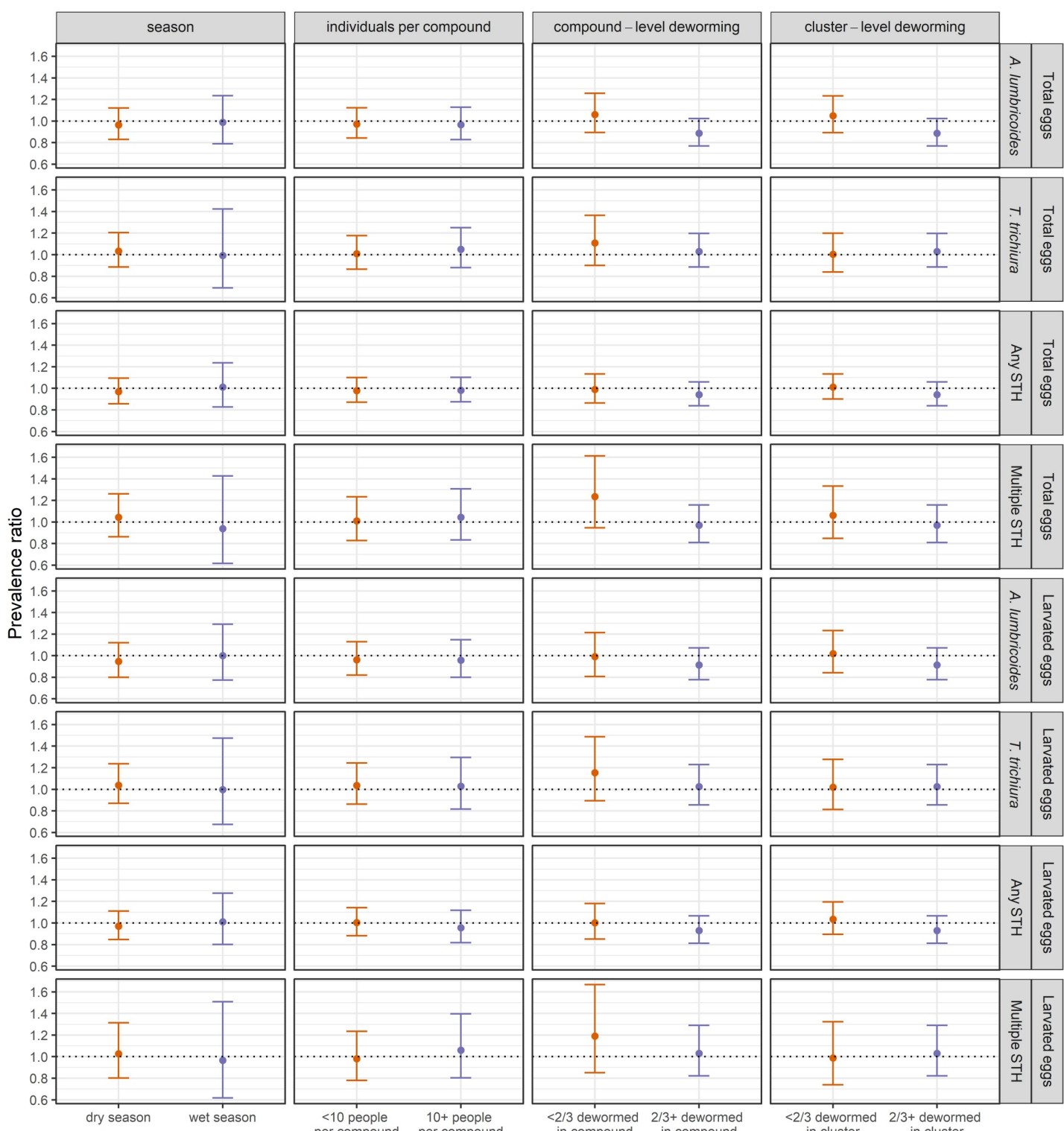

**Fig 3. Prevalence ratio for total and larvated *A. lumbricoides* eggs, *T. trichiura* eggs, any soil-transmitted helminth eggs, and multiple species of soil-transmitted helminth eggs by subgroup.**

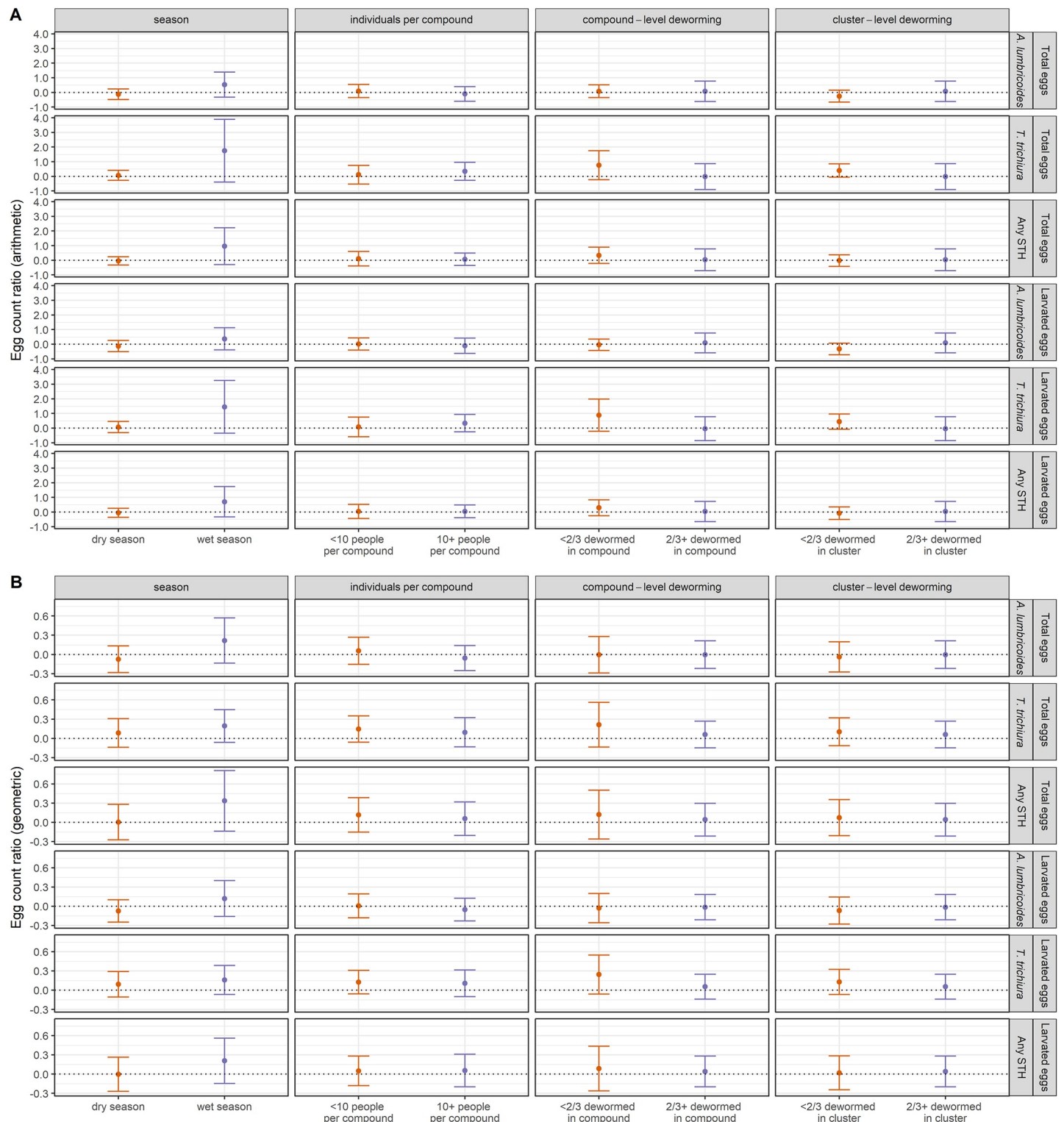

**Fig 4. Arithmetic and geometric egg count ratios for total and larvated *A. lumbricoides* eggs, *T. trichiura* eggs, any soil-transmitted helminth eggs, and multiple species of soil-transmitted helminth eggs by subgroup.**

*T. trichiura* [16]. In Kenya, 23% of enrolled children were infected with *A. lumbricoides* and 1% with *T. trichiura*, with the geometric mean fecal egg count of 4.0 epg for *A. lumbricoides* and 0.5 epg for *T. trichiura* [14]. Differences in ambient and soil temperature, relative humidity, soil moisture and soil texture may also have increased the survival of STH eggs in the soil in Bangladesh compared to Kenya. Mean soil moisture during the study was 15.8% in Bangladesh vs. 11.0% in Kenya [18].

Use of improved sanitation facilities would be expected to reduce environmental contamination with STH eggs by reducing open defecation and isolating feces from the environment until STH eggs have degraded. Given that approximately 30% of children <2 years old in the control and sanitation arms were infected with *A. lumbricoides* and 3–5% were infected with *T. trichiura* in our study population, continuing child open defecation and unsafe disposal of child feces in the sanitation arm could be a source of STH eggs to contaminate courtyards and contribute to the lack of impact on STH eggs in the environment from the sanitation intervention.

Another reason for why the prevalence and concentration of *A. lumbricoides* and *T. trichiura* eggs was not significantly lower in the sanitation arm compared to the control arm could be that Bangladeshi households periodically build up the height of their courtyard and household floor using agricultural soil. This soil may have been contaminated by non-study households that lived in the same community as study households. The WASH Benefits trial only enrolled approximately 10% of households in a given geographical area (due to the trial's eligibility criteria), and enrolled households were therefore surrounded by non-enrolled households that continued their day-to-day sanitation practices. Their sanitation practices may have contaminated the soil with soil-transmitted helminths. STH-contaminated soil may also have been carried on the soles of household members' shoes or feet back to the courtyard soil of study households, but this practice on its own is unlikely to account for the high STH levels observed.

Domestic animals are known reservoir hosts for some human-infecting STH (*A. lumbricoides* in pigs, *A. ceylanicum* in dogs and cats). These domestic animals are uncommon in rural Bangladesh. However, the domestic animals that are present may distribute STH eggs across household courtyards by passage after coprophagia. In the control arm, 91% of households had chickens, 69% had cows, and 39% had goats and/or sheep, and 56% households allows their animals to roam freely [48]. These animals may step on feces in the courtyard, in neighboring compounds, in agricultural fields or patches of bush, and then track STH eggs on their feet and hooves back to the courtyard where they stay. Additionally, animals may intentionally or inadvertently consume feces contaminated with STH eggs or food contaminated with soil that contains STH eggs and deposit these eggs in their own feces. After chickens were fed viable porcine *A. lumbricoides* (formerly known as *A. suum*) [49] and *T. suis* eggs, these eggs were recovered in their feces and, after embryonation in vermiculite, were able to cause infection in pigs [50]. The ten chickens studied passed through 61% of porcine *A. lumbricoides* and 42% of *T. suis* eggs they ingested. Twelve days after pigs had been inoculated with embryonated eggs that had passed through the chickens, 34% of porcine *A. lumbricoides* eggs and 82% of *T. suis* eggs were recovered from the pig feces [50]. This suggests that chickens may act as transfer hosts for STH species that are infective to humans. Animal feces were observed in 85% of sanitation compounds and 92% of control compounds [34], indicating that animal feces, in addition to human feces, may contribute to the re-inoculation of soil with STH eggs. In future studies, use of molecular detection methods could distinguish STH species that commonly infect animals from those that more commonly infect humans.

Community-level, sustained sanitation and/or animal confinement interventions could prevent community-wide STH-contamination of soil, thereby limiting exposure to STH ova

and larvae. We are not aware of any studies that have examined the impact of community-wide animal confinement interventions on STH in soil or STH infections in children. With regards to community-level human feces management, previous work has found mixed evidence on whether increased access to latrines is associated with reduced STH infection rates. An observational study in rural Ethiopia found that, after adjusting for individual, household, and community characteristics, hookworm prevalence was not associated with community-level latrine usage (defined as the proportion of households in a cluster that had observed indicators of latrine use such as a defined path to a latrine and feces observed in the latrine pit) [51]. In the same study, *T. trichiura* prevalence and *A. lumbricoides* prevalence were higher in communities where community latrine usage was ≥60% *vs.* <20%. Household latrine usage did not modify the effect of community latrine usage for hookworm or *T. trichiura* prevalence. In communities with latrine usage ≥80%, children in households with a latrine in use had a significantly lower prevalence of *A. lumbricoides* compared to children in households without a latrine. In contrast, in communities with sanitation coverage <20%, children in households with a latrine in use had a significantly *higher* prevalence of *A. lumbricoides* compared to children in households without a latrine. The authors posited that higher community latrine usage may be associated with urbanization, which has been associated with increased STH prevalence and may not have been fully accounted for by including population density in their models. They also suggested that *A. lumbricoides* prevalence in communities with low coverage of latrines may be driven by increased latrine sharing. In India, a cluster-randomized trial of latrine promotion and construction found that 46% of people in the intervention arm had access to a functional latrine at the end of the trial compared to 15% of people in the control arm. Despite this, the intervention had no effect on the prevalence or intensity of hookworm, *A. lumbricoides*, or *T. trichiura* infections [11]. In an observational study in Vietnam, 98% of households had latrines, but infection with STH was 58% for hookworm, 45% for *T. trichiura*, and 14% for *A. lumbricoides* [52]. One hypothesis for this finding was that households in this community use fresh night soil as fertilizer. These finding suggest that the impact of community-wide sanitation may be mediated by factors such as household latrine use, latrine sharing, and use of human fecal waste for fertilization, and confounded by factors such as urbanization.

Our finding that the sanitation intervention did not significantly reduce prevalence or concentration of STH eggs in courtyard soil compared to the control arm is consistent with findings from the WASH Benefits trial that the prevalence of *A. lumbricoides* infection in children was not significantly different in the sanitation *vs.* control arms [16]. This could indicate that persistent *A. lumbricoides* eggs in soil continued to infect children in the sanitation arm, that infected children continued to contaminate the soil in the sanitation arm, or a combination of both. In contrast, the WASH Benefits trial found a 29% reduction in *T. trichiura* infection in children in the sanitation vs. control arms. However, this could be a chance finding as there was no significant reduction of *T. trichiura* infection in the other trial arms that included a sanitation component [16]. Our findings of no difference in the prevalence or concentration of *T. trichiura* in soil between the sanitation and control arms supports the possibility that the observed reduction in *T. trichiura* infection in the sanitation arm could be a chance finding. Our findings are also consistent with other environmental measurements conducted among trial households showing no reduction in *E. coli* contamination or bacterial, viral and protozoan pathogens in soil (or other types of environmental samples) between the sanitation and control arms [53,54]. A parallel study nested within the WASH Benefits Kenya trial also found no reduction in STH eggs in soil from the sanitation intervention [18]. Consistent with the theory that water, sanitation, and hygiene interventions are more effective when coupled with mass drug administration [13], our pre-specified subgroup analyses showed that the sanitation intervention had a modest reduction effect on STH eggs in soil in clusters and compounds in

which ≥2/3 of children were dewormed. We cannot rule out the role of chance in these findings as the effects were borderline and multiple subgroup analyses increase the likelihood of chance findings.

One limitation of this study is that we were unable to recover hookworm eggs from the soil and therefore cannot draw conclusions about their presence in the environment. The WASH Benefits trial found a borderline 24% reduction in hookworm infection in children in the sanitation arm [16]; we could not assess the effect of the sanitation intervention on hookworm eggs in soil as we did not detect any. Detection of hookworm eggs in soil has been reported in only two studies; of 345 total samples in these studies, hookworm eggs were detected in three samples [27,28]. The inability of other studies to detect hookworm eggs in soil indicates that these three samples may have had eggs of other species misclassified as hookworm. It is possible that hookworm eggs degraded during our sample processing steps. In order to assess the prevalence and concentration of hookworm in soil, hookworm larvae (rather than eggs) can be recovered from soil samples. One study successfully used damp gauze pads placed on the soil surface to recover hookworm larvae [21]. A related limitation is that our method was optimized for the recovery of *A. lumbricoides* eggs from soil [41] and the recovery efficiency of *T. trichiura* eggs is unknown. While this does not affect the estimated effect of the sanitation intervention on *T. trichiura* eggs in soil, it influences the interpretation of the prevalence and concentration of *T. trichiura* we observed in soil. Finally, we relied on microscopy to quantify STH eggs. Microscopic inspection is vulnerable to misclassification of STH eggs [55,56], and has lower sensitivity and specificity compared to DNA-based diagnostics [57,58]. Future work to validate DNA-based methods for detection of STH in soil could improve the reliability of future studies and allow for inclusion of more fragile STH species, such as hookworm.

A compound-level sanitation intervention consisting of lined pit latrines, child potties and scoops for feces disposal was not effective in reducing the prevalence and concentration of STH eggs in courtyard soil in rural Bangladesh. Access to and use of latrines by adults was insufficient to prevent feces from contaminating the domestic environment when the majority of young children continued to practice open defecation in the household courtyard and the feces of children and animals were not disposed of hygienically. In order to effectively reduce the transmission of STH infections in low-resource contexts, interventions may need to focus on widespread access to and use of hardware for hygienic disposal of child and animal feces in addition to adult feces.

## Supporting information

**S1 Appendix. Effect of sanitation improvements on soil-transmitted helminth eggs in courtyard soil from rural Bangladesh: Evidence from a cluster-randomized controlled trial.** Table A: Unadjusted and adjusted prevalence ratio and prevalence difference for sanitation intervention vs. control arm for all soil-transmitted helminths and larvated soil-transmitted helminths. Table B: Unadjusted and adjusted egg count reduction for sanitation intervention vs. control arm for all soil-transmitted helminths and larvated soil-transmitted helminths. Table C: Prevalence of all soil-transmitted helminths in control arm, by subgroup. Table D: Concentration of all soil-transmitted helminths in control arm, by subgroup. Table E: Unadjusted prevalence ratio and prevalence difference for sanitation intervention vs. control arm for soil-transmitted helminth eggs, by subgroups. Table F: Unadjusted prevalence ratio and prevalence difference for sanitation intervention vs. control arm for larvated soil-transmitted helminth eggs, by subgroups. Table G: Unadjusted egg count reduction for sanitation intervention vs. control arm for soil-transmitted helminth eggs, by subgroups. Table H: Unadjusted egg count reduction for sanitation intervention vs. control arm for larvated soil-

transmitted helminth eggs, by subgroups. Fig A: Prevalence difference for total and larvated A. lumbricoides eggs, T. trichiura eggs, any soil-transmitted helminth eggs, and multiple species of soil-transmitted helminth eggs, by subgroup.
(DOCX)

## Acknowledgments

We gratefully acknowledge the WASH Benefits Bangladesh study families who participated in this study and provided environmental samples.

## Author Contributions

**Conceptualization:** Ayse Ercumen.

**Data curation:** Ayse Ercumen.

**Formal analysis:** Laura H. Kwong, Benjamin F. Arnold, Alan Hubbard, Ayse Ercumen.

**Funding acquisition:** Alan Hubbard, Leanne Unicomb, John M. Colford, Jr., Stephen P. Luby, Ayse Ercumen.

**Investigation:** Debashis Sen, Sharmin Islam, Sunny Shahriar.

**Methodology:** Laura H. Kwong, Debashis Sen, Sharmin Islam, Kara Nelson.

**Project administration:** Sarker Masud Parvez, Mahfuza Islam, Leanne Unicomb, Md. Mahbubur Rahman.

**Supervision:** Ayse Ercumen.

**Visualization:** Laura H. Kwong.

**Writing – original draft:** Laura H. Kwong.

**Writing – review & editing:** Laura H. Kwong, Debashis Sen, Sharmin Islam, Sunny Shahriar, Jade Benjamin-Chung, Benjamin F. Arnold, Alan Hubbard, Sarker Masud Parvez, Leanne Unicomb, Md. Mahbubur Rahman, Kara Nelson, John M. Colford, Jr., Stephen P. Luby, Ayse Ercumen.

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
