## [Decision Letter · Decision Letter 0]

7 Jan 2021

Dear Dr. Kwong,

Thank you very much for submitting your manuscript "Effect of sanitation improvements on soil-transmitted helminth eggs in courtyard soil from rural Bangladesh: Evidence from a cluster-randomized controlled trial" for consideration at PLOS Neglected Tropical Diseases. As with all papers reviewed by the journal, your manuscript was reviewed by members of the editorial board and by several independent reviewers. The reviewers appreciated the attention to an important topic. Based on the reviews, we are likely to accept this manuscript for publication, providing that you modify the manuscript according to the review recommendations. 

Please carefully address the reviewer's comments before submitting your revision.

Sincerely,

Richard Stewart Bradbury, PhD

Associate Editor

Suzy Campbell

Deputy Editor

Please carefully address the reviewer's comments before submitting your revision.

Reviewer's Responses to Questions

**Key Review Criteria Required for Acceptance?**

**Methods**

-Are the objectives of the study clearly articulated with a clear testable hypothesis stated?

-Is the study design appropriate to address the stated objectives?

-Is the population clearly described and appropriate for the hypothesis being tested?

-Is the sample size sufficient to ensure adequate power to address the hypothesis being tested?

-Were correct statistical analysis used to support conclusions?

-Are there concerns about ethical or regulatory requirements being met?

Reviewer #1: The authors sufficiently describe the study objectives and design. The design is appropriate, high quality, and well-described in this manuscript. Further details on the WASH Benefits study are available elsewhere. The population is clearly described, and the sample size is appropriate. The statistical analysis is appropriate. I have no concerns about ethical issues.

Reviewer #2: • This study is a substudy as part of the larger WASH Benefits clinical trial, and as such the main study population is already defined.

• The authors clearly state the aim of this paper – to assess the impact of a compound sanitation intervention on the prevalence and abundance of several different species of STH in soil obtained from outside control and intervention households

• The study design is appropriate, though I would like to see mention of (and ideally inclusion of) additional potential confounders in this study. The authors state that they are primarily interested in unadjusted effects, and state that “as randomisation led to a good balance across arms, bias due to confounding in our dataset is unlikely”. However, the authors mention in the introduction that 94% of adults in the intervention group of the main trial use latrines provided in the intervention, compared with 40% in the control group (though I note it is unclear if this percentage means 40% of control households that already have a hygienic toilet actually use it, or 40% of the whole control population actually have a hygienic toilet). They also report that open defecation is still a problem in both arms, with high percentages of children still practicing OD (e.g. 81% of children aged 0 – 2 years). They also mention that unhygienic disposal of child faeces is also common, with 84% of control and 64% of intervention households not safely disposing of child faeces. However, none of these factors are measured or adjusted for in the analysis, and I am concerned they may be unmeasured confounders. They seem likely predictors of STH infection of soil in a household compound. Were these factors measured in this subanalysis? If so they need to be adjusted for, and if not, then they need to be mentioned as a limitation or possible explanation for why there is no difference between control/intervention

• The methodology (and introduction) would benefit from a few more definitions. For example, defining what “abundance” means (eggs per gram of soil), and interpreting ICC values e.g. excellent/good/moderate/fair/poor to help the reader. Also, a brief explanation for how prevalence ratios are calculated. They mention “randomisation leading to a good balance”, but would be useful to add “a good balance of sociodemographic characteristics” and add in a reference to table 1.

• The authors do not provide detail about how the analysis is carried out in R. They mention and reference use of R – “We estimated unadjusted and adjusted intention-to-treat effects using the R (version 3.3.2) package for targeted maximum likelihood estimation with SuperLearner (42,43)”, but no additional information is given. This makes it very difficult for the reader to understand how the analysis was conducted, especially adjusting for potential confounders

• The data collection also took place over a year (May 2015 – May 2016), which is a long period of time, though I assume this is because of the need to include wet and dry seasons. It would be good to see “time since intervention” included as a measured factor, as I’m not sure how fieldwork was planned/enacted over this long period of time, and if there were different lengths of time between intervention enactment and soil sample analysis (e.g. 12 months for one household, vs, 24 months for others - there could be a significant influence of time between intervention and analysis that could impact the results) 

• Authors mention carrying out “replicate” samples, but would be good to state if duplicate or triplicate. The authors mention “Replicates were dropped from the analysis”, but do not say if they were averaged first across replicates, or how they decided which to drop – would be good to explain this.

• Authors reference the clinical trial number that this substudy is nested under, but do not mention ethical approval – this should be stated to ensure the study has been approved

**Results**

-Does the analysis presented match the analysis plan?

-Are the results clearly and completely presented?

-Are the figures (Tables, Images) of sufficient quality for clarity?

Reviewer #1: The analysis is appropriate and well described. The results are sufficiently and clearly presented. The figures are clear and appropriate.

Reviewer #2: • Overall the results demonstrate that there was no significant impact of the intervention on STH prevalence or abundance

• The analyses carried out support the research question outlined, though as noted above, it is difficult to understand how the analysis is conducted with the limited information provided about the R package used (without digging around to find out more yourself).

• The authors should consider if there is a better way to convey the information in the tables and graphs more concisely - they are difficult to navigate as a reader as there is so much information presented

 o I would recommend that the individual graphs in figure 1 are combined into a single graph rather than 8 individual graphs that have the same axes; I would make the same recommendation for figure 2 and 3 – this would also help the reader to compare the different results between species more easily as well as reducing the number of individual graphs

 o For tables 2 – 5, I would recommend moving substantial amounts of this data into supplementary figures, and choose only the most important information to include in the main paper. This is a lot of information to include in the main paper, and I don’t feel it adds value to include it all. The authors should decide which are the most important table values to include – some suggestions could be to choose to show “any eggs” results in the main paper and “larvated eggs” results in the supplementary figures? Or just show adjusted figures in the main paper and unadjusted in the supplementary (though I realise the primary parameters chosen are unadjusted so this might not work). Or just show unadjusted overall figures in the main paper and move the sub-analyses results to the supplementary figures. There are a few ways to do it – the authors should decide which are the most important findings to show in the main paper and move the rest to supplementary

• It was also difficult to compare adjust vs unadjusted values – adjusted were given as supplementary and unadjusted were in the main tables. 

• Again would be good to help the reader by interpreting the ICC values

• In the subgroup analyses the authors make a number of statements about differences in STH egg prevalence/intensity values in the dry vs. wet season, and use the word “significant”. The authors need to add a brief explanation/justification as to how they are deciding a finding is significant or not, as there doesn’t seem to be justification/evidence/calculations of how some differences are judged significant and others are not e.g. p values

• In the final paragraph of the results section the authors mention modest reductions in prevalence of ascaris in the deworming subgroup analysis – again, it would be good for the authors to expand on how they are determining what is considered a real statistically significant difference and how this is justified - is it just the confidence intervals they are using in this instance?

**Conclusions**

-Are the conclusions supported by the data presented?

-Are the limitations of analysis clearly described?

-Do the authors discuss how these data can be helpful to advance our understanding of the topic under study?

-Is public health relevance addressed?

Reviewer #1: The conclusions are supported by the data presented and some limitations are addressed. The authors address the public health relevance of the results.

Reviewer #2: • The authors provide a number of interesting and reasonable arguments to support why no significant difference was observed between control and intervention arms for a number of parameters. For example, how it was common to top up soil in courtyards (and so STH could be introduced this way), or how household members could walk STH into the courtyard from other nearby soil on their shoes

• In the opening paragraph of the discussion, the authors mention hygienic latrine use and disposal of child faeces in the intervention arm (“Among households in the sanitation arm, where adults regularly used hygienic latrines but only a third of households safely disposed of child feces the prevalence or abundance of total or larvated A. lumbricoides or T. trichiura eggs in courtyard soil was not significantly lower compared to households in the control arm”). These factors do not appear to have been measured or adjusted for in this substudy analysis. The authors do at least touch on how “continuing child open defecation and unsafe disposal of child feces in the sanitation arm could be a source of STH eggs to contaminate courtyards and contribute to the lack of impact on STH eggs in the environment from the sanitation intervention”, but it would be good to make this clearer and more obvious as a limitation in this study (as it doesn’t seem to have been measured or adjusted for, and would potentially be a strong risk factor)

• I am again uncomfortable about mention of “significant differences” without justification in the results section

• The authors mention future use of qPCR to detect STH (especially hookworm, which they weren’t able to detect using their methodology) but I think this could be a stronger recommendation with authors recommending urgent work to validate this method (which is successfully used to detect STH species from faecal samples already) in soil samples.

**Editorial and Data Presentation Modifications?**

Reviewer #1: Line 51: "isolate and hygienically dispose of" - should include "adults"

Introduction section - Suggest inclusion of some information about the Bangladesh context; paper generally reflects little if anything about STH epidemiology and STH control in Bangladesh - that's to the detriment of the manuscript in my opinion. 

Line 75: Per previous comment - Bangladesh WHO recommends "annual" MDA but Bangladesh treats school-attending school-age children 2x annually, with little documented coverage of non-school attending SAC or preschool age children. Those issues are relevant to your results (including around young children). 

Line 97: unclear language

Lines 107-9: Suggest moving this sentence to paragraph starting on Line 82

Lines 123-4: "Eight nearby pregnant..." You mean households with pregnant women? Presumably, you mean nearby to one another.

Line 143: "structured observations" - observed by who, when? 

Line 191: remove "quality assurance." Confirmatory testing would be considered, quality control. 

Line 228: "...analyses accounted for [the] lab staff member who..."

Line 241 & 243: would remove the word, "status" - confusing

Discussion: Discussion is focused and clearly presented. And I would suggest consideration and discussion of the context of the study even though that context is outside of WASHBenefits - it speaks directly to implications and public health relevance. Regarding the comparison with Kenya - the contexts are quite different as you touch on. However, you don't discuss preventive chemotherapy coverage across the entire pop in each country or baseline prevalence. 

Lines 335-7: Excellent point and one that I think is central to the broader relevance of the work - but unfortunately not discussed. 

Line 339-42: You intentionally sampled one small, specific, high-traffic (in front of main entrance to household) area across all HHs, (how) does that affect results? 

Lines 407-409: See comments from lines 335-7. Suggest that instead of leaving this for the last sentence, a more thorough discussion be had, and evidence (or the lack thereof) from other RCTs/studies of this quality be raised.

Reviewer #2: The main recommendation would be to modify the figures and tables significantly so they are easier to interpret and understand. It may be better for the authors to decide their priorities rather than an editor given there is a lot of data. For example, combine the individual graphs into one for figures 1/2/3, and pick the most important information to present from the tables (rather than presenting everything).

There are a few minor editorial issues I noticed e.g. “STH eggs in soil have been detected in outdoor soil in various settings” [don’t need to say in soil twice]; “As the method was optimized to detect A. lumbricoides, the recovery efficiency for other STH species was likely was lower” [remove one of the “was” occurrences]; “Based on the daily processing capacity of the lab, we targeted to sample approximately 1500 compounds” [we “aimed/planned/set out to sample” may be better English] etc.

**Summary and General Comments**

Reviewer #1: Well-executed study and well-written manuscript. Data analysis is appropriate and well presented. While beyond the immediate aim of WASH Benefits, I would encourage you to more thoroughly discuss the broader relevance of your findings (e.g. need for sustained community-level WASH interventions instead of HH-level interventions, etc.). Obviously, there are practical challenges (e.g. cost) with implementing a similar RCT at community-level.

Reviewer #2: This paper was interesting, and I believe important findings and ideas are presented that will be very interesting to others working in this area.

In general, the details in the methods and results need tightening up, with additional definitions/clarifications added to avoid confusion or misunderstanding by the reader. For example, an explanation for how significance is assessed; acknowledgement of unmeasured confounders (such as household behaviour with regard to sanitation usage and child faeces disposal practices) is also important – is this information available and could it be included in the adjusted analysis?; and more information about how the analysis was carried out (what the R package does).

My recommendation would be to revise all of the tables and figures (excluding the sociodemographics table, which is fine) to make them easier to navigate, and a lot more concise, and by moving less important information to the supplementary figure section. 

I believe this paper would be suitable for publication after a moderate revision. Minor revisions are needed to the text (e.g. additional clarifications), and some more major adjustments to the figures/tables.

PLOS authors have the option to publish the peer review history of their article (what does this mean?). If published, this will include your full peer review and any attached files.

Reviewer #1: No

Reviewer #2: Yes: Dr Clare E F Dyer
---

## [Editor Report · Decision Letter 1]

19 Apr 2021

Dear Dr. Kwong,

Thank you very much for submitting your manuscript "Effect of sanitation improvements on soil-transmitted helminth eggs in courtyard soil from rural Bangladesh: Evidence from a cluster-randomized controlled trial" for consideration at PLOS Neglected Tropical Diseases. As with all papers reviewed by the journal, your manuscript was reviewed by members of the editorial board and by several independent reviewers. The reviewers appreciated the attention to an important topic. Based on the reviews, we are likely to accept this manuscript for publication, providing that you modify the manuscript according to the review recommendations. 

The authors have addressed all of the reviewer's comments from the first submission. In my editorial review, I have noted some errors or misunderstandings which need to be addressed before this article may be accepted. These are detailed below:

Lines 71 & 72: Ancylostoma ceylanicum should now be included in any list of the human hookworms. It is the second most common hookworm of humans in the Asia/Pacific region, with high prevalence in some regions.

Lines 155-157: What exact month or months of what year were the soil samples examined in this study collected? The degree of rainfall at the time of collection will be very pertinent to results and should be made available to readers.

Lines 197, 202, 208: Please add the manufacturer and city of manufacture for 7X detergent solution, zinc sulfate and sulfuric acid.

Lines 227-228. Note also because of the markedly different life cycles of this STH meaning that eggs are less likely to be present in any tested patch of soil [Hookworm larvae hatch after 48-72 hours and seek moisture, so you see them in moisture areas (particularly dewy grass) close to deposited faeces, whereas Trichuris and Ascaris eggs remain viable in the soil for up to three years or more. The hookworm larvbae only live for two or three months in the environment. So, you don’t have years worth of hookworm eggs floating around in dust after desiccation of the faeces, to be picked up on hands and shoes and brought into households, as you do with Trichuris and Ascaris].

Lines 386-402: I assume that household dogs are not present in Bangladesh? These provide a reservoir of A. ceylanicum hookworm and also disseminate T. trichiura and A. lumbricoides from human faeces via coprophagia. What about cats? They are also a reservoir of A. ceylanicum.

A. Suum has been shown to be a at most a sub-genotype of A. lumbricoides and more likely the same species, just in a different host. Pigs may often be infected with the human genotype of A. lumbricoides and vice versa, particularly in LIMC settings. Please change any reference to A. suum to A. lumbricoides. Also note that change line 386, which is incorrect, pigs may be a host for human A. lumbricoides. You can use the terms “porcine A. lumbricoides” or “porcine ascariasis” if you need to differentiate human from pig infections. Please see links to references supporting this below:

https://academic.oup.com/jid/article/213/8/1355/2459518?login=true

https://parasitesandvectors.biomedcentral.com/articles/10.1186/1756-3305-5-42

https://www.sciencedirect.com/science/article/pii/S0378111911006494

https://www.ncbi.nlm.nih.gov/pmc/articles/PMC4799673/

https://bmcvetres.biomedcentral.com/articles/10.1186/1746-6148-10-99

Sincerely,

Richard Stewart Bradbury, PhD

Associate Editor

Suzy Campbell

Deputy Editor

The authors have addressed all of the reviewer's comments from the first submission. In my editorial review, I have noted some errors or misunderstandings which need to be addressed before this article may be accepted. These are detailed below:

Lines 71 & 72: Ancylostoma ceylanicum should now be included in any list of the human hookworms. It is the second most common hookworm of humans in the Asia/Pacific region, with high prevalence in some regions.

Lines 155-157: What exact month or months of what year were the soil samples examined in this study collected? The degree of rainfall at the time of collection will be very pertinent to results and should be made available to readers.

Lines 197, 202, 208: Please add the manufacturer and city of manufacture for 7X detergent solution, zinc sulfate and sulfuric acid.

Lines 227-228. Note also because of the markedly different life cycles of this STH meaning that eggs are less likely to be present in any tested patch of soil [Hookworm larvae hatch after 48-72 hours and seek moisture, so you see them in moisture areas (particularly dewy grass) close to deposited faeces, whereas Trichuris and Ascaris eggs remain viable in the soil for up to three years or more. The hookworm larvbae only live for two or three months in the environment. So, you don’t have years worth of hookworm eggs floating around in dust after desiccation of the faeces, to be picked up on hands and shoes and brought into households, as you do with Trichuris and Ascaris].

Lines 386-402: I assume that household dogs are not present in Bangladesh? These provide a reservoir of A. ceylanicum hookworm and also disseminate T. trichiura and A. lumbricoides from human faeces via coprophagia. What about cats? They are also a reservoir of A. ceylanicum.

A. Suum has been shown to be a at most a sub-genotype of A. lumbricoides and more likely the same species, just in a different host. Pigs may often be infected with the human genotype of A. lumbricoides and vice versa, particularly in LIMC settings. Please change any reference to A. suum to A. lumbricoides. Also note that change line 386, which is incorrect, pigs may be a host for human A. lumbricoides. You can use the terms “porcine A. lumbricoides” or “porcine ascariasis” if you need to differentiate human from pig infections. Please see links to references supporting this below:

https://academic.oup.com/jid/article/213/8/1355/2459518?login=true

https://parasitesandvectors.biomedcentral.com/articles/10.1186/1756-3305-5-42

https://www.sciencedirect.com/science/article/pii/S0378111911006494

https://www.ncbi.nlm.nih.gov/pmc/articles/PMC4799673/

https://bmcvetres.biomedcentral.com/articles/10.1186/1746-6148-10-99

Figure Files:

Data Requirements:

Reproducibility:

References

---

## [Decision Letter · Decision Letter 2]

18 Jun 2021

Dear Dr. Kwong,

Thank you very much for submitting your manuscript "Effect of sanitation improvements on soil-transmitted helminth eggs in courtyard soil from rural Bangladesh: Evidence from a cluster-randomized controlled trial" for consideration at PLOS Neglected Tropical Diseases. As with all papers reviewed by the journal, your manuscript was reviewed by members of the editorial board and by several independent reviewers. The reviewers appreciated the attention to an important topic. Based on the reviews, we are likely to accept this manuscript for publication, providing that you modify the manuscript according to the review recommendations. 

This paper is now nearing a state where it might be accepted. The authors are encouraged to revise the paper in line with the further reviewer comments.

Sincerely,

Richard Stewart Bradbury, PhD

Associate Editor

Suzy Campbell

Deputy Editor

This paper is now nearing a state where it might be accepted. The authors are encouraged to revise the paper in line with the further reviewer comments.

Reviewer's Responses to Questions

**Key Review Criteria Required for Acceptance?**

**Methods**

-Are the objectives of the study clearly articulated with a clear testable hypothesis stated?

-Is the study design appropriate to address the stated objectives?

-Is the population clearly described and appropriate for the hypothesis being tested?

-Is the sample size sufficient to ensure adequate power to address the hypothesis being tested?

-Were correct statistical analysis used to support conclusions?

-Are there concerns about ethical or regulatory requirements being met?

Reviewer #3: (No Response)

**Results**

-Does the analysis presented match the analysis plan?

-Are the results clearly and completely presented?

-Are the figures (Tables, Images) of sufficient quality for clarity?

Reviewer #3: (No Response)

**Conclusions**

-Are the conclusions supported by the data presented?

-Are the limitations of analysis clearly described?

-Do the authors discuss how these data can be helpful to advance our understanding of the topic under study?

-Is public health relevance addressed?

Reviewer #3: (No Response)

**Editorial and Data Presentation Modifications?**

Reviewer #3: (No Response)

**Summary and General Comments**

Reviewer #3: I would like to thank the authors for consideration of, and revisions in response to, my comments. Unfortunately some of the corrected statements require some rewriting for clarificiation as in their current form they might confuse the literature. The identified issues and some suggested corrections are listed below. 

Lines 294-295:

I think that it is important to clarify which species of STH may cause patent infection in these animals. Currently, these lines read as,

"Few households had animals that can spread STH by consuming human feces (pigs, dogs) and/or are known hosts for the STH species we investigated (pigs, dogs, cats)."

I'd suggest the following potential change,

""Few households had animals that can spread STH by passage after consuming human feces (pigs, dogs) and/or are known hosts for the STH species we investigated (A. lumbricoides in pigs, A. ceylanicum in dogs and cats)."

Lines 392-394:

"Domestic animals that consume human feces (pig and dogs) or are known hosts for Ascaris spp. (pigs, dogs, and cats) are uncommon in rural Bangladesh. However, the domestic animals that are present may distribute STH eggs across household courtyards."

- Cats and dogs are patent hosts for A. ceylanicum, but not hosts for Ascaris (though this is has been suggested for dogs once or twice in the literature, it has not yet been proven). Pigs are true patent hosts for Ascaris.

Suggested alternative-

"Domestic animals are known reservoir hosts for some human-infecting STH (A. lumbricoides in pigs, A. ceylanicum in dogs and cats). These domestic animals are uncommon in rural Bangladesh. However, the domestic animals that are present may distribute STH eggs across household courtyards by passage after coprophagia."

Following these comments being addressed, I have no further revisions.

PLOS authors have the option to publish the peer review history of their article (what does this mean?). If published, this will include your full peer review and any attached files.

Reviewer #3: Yes: Richard Bradbury

Figure Files:

Data Requirements:

Reproducibility:

References

---

## [Editor Report · Decision Letter 3]

24 Jun 2021

Dear Dr. Kwong,

We are pleased to inform you that your manuscript 'Effect of sanitation improvements on soil-transmitted helminth eggs in courtyard soil from rural Bangladesh: Evidence from a cluster-randomized controlled trial' has been provisionally accepted for publication in PLOS Neglected Tropical Diseases.

Best regards,

Richard Stewart Bradbury, PhD

Associate Editor

Suzy Campbell

Deputy Editor

---

## [Editor Report · Acceptance letter]

21 Jul 2021

Dear Dr. Kwong,

We are delighted to inform you that your manuscript, "Effect of sanitation improvements on soil-transmitted helminth eggs in courtyard soil from rural Bangladesh: Evidence from a cluster-randomized controlled trial," has been formally accepted for publication in PLOS Neglected Tropical Diseases.

Best regards,

Shaden Kamhawi

co-Editor-in-Chief

Paul Brindley

co-Editor-in-Chief
